# *Pterospartum tridentatum* Liqueur Using Spirits Aged with Almond Shells: Chemical Characterization and Phenolic Profile

**DOI:** 10.3390/molecules28114455

**Published:** 2023-05-31

**Authors:** Cátia Garcia, Maria Inês Dias, Marta H. F. Henriques, Lillian Barros, Fernando Ramos

**Affiliations:** 1Faculty of Pharmacy, University of Coimbra, Azinhaga de Santa Comba, 3000-548 Coimbra, Portugal; catia.garcia.97@gmail.com; 2Centro de Investigação de Montanha (CIMO), Instituto Politécnico de Bragança, Campus de Santa Apolónia, 5300-253 Bragança, Portugal; maria.ines@ipb.pt; 3Laboratório Associado para a Sustentabilidade e Tecnologia em Regiões de Montanha (SusTEC), Instituto Politécnico de Bragança, Campus de Santa Apolónia, 5300-253 Bragança, Portugal; 4Polytechnic Institute of Coimbra, Coimbra Agriculture School, Bencanta, 3045-601 Coimbra, Portugal; mhenriques@esac.pt; 5Research Centre for Natural Resources Environment and Society (CERNAS), Coimbra Agriculture School, Bencanta, 3045-601 Coimbra, Portugal; 6Associated Laboratory for Green Chemistry (LAQV) of the Network of Chemistry and Technology (REQUIMTE), Rua D. Manuel II, Apartado, 55142 Porto, Portugal

**Keywords:** *Pterospartum tridentatum*, liqueur, physicochemical characteristics, HPLC-DAD/ESI-MSn, sensory analysis

## Abstract

With great cultural significance, spirits and distillate beverages constitute an important niche market in Europe. The development of new food products, particularly for the functionalization of these beverages, is increasing exponentially. The present work aimed to develop a new wine spirit beverage aged with almond shells and flowers of *P. tridentatum* for further characterization of bioactive and phenolic compounds, coupled with a sensorial study to evaluate the acceptance of this new product by the market. Twenty-one phenolic compounds were identified, mainly isoflavonoids and *O*- and *C*-glycosylated flavonoids, especially in *P. tridentatum* flowers, indicating that it is a highly aromatizing agent. The developed liqueur and wine spirits (almonds and flowers) showed distinct physicochemical properties, with the last two samples showing greater appreciation and purchase intention by consumers due to their sweetness and smoothness. The most promising results were found for the *carqueja* flower, which should be further investigated in an industrial context to contribute to its valorization in its regions of origin, such as Beira Interior and Trás-os-Montes (Portugal).

## 1. Introduction

*Pterospartum tridentatum*, known as *carqueja* in Portugal, is a shrub that grows spontaneously in the mountains of Portugal. Its flowers, known for their medicinal properties, blossom in spring and are commonly used to prepare teas and infusions and as a seasoning agent for rice or meat dishes [1,2,3]. This plant does not present toxicity to the consumer at doses below 100 mg/kg [1] and is commonly used in the treatment of sore throats, diabetes, hypertension and hypercholesterolemia and for producing a diuretic effect [1,3,4,5]. It is known to have high anti-inflammatory properties related to its antioxidant activity and high content of phenolic compounds, which fight free radicals that originate during oxidative stress [1,5]. Several studies showed that some of the main phenolic compounds found in aqueous and ethanolic extracts of *Pterospartum tridentatum* are flavonoids, specifically, luteolins, taxifolin, genestin, genestein, prunetin, quercetins and biochanin A, but also come phenolic acids, namely, rosmarinic acid [3,5,6,7]. Previous reports [1,8] have also stated a high content of tocopherols, sugars (with fructose being the most relevant), organic acids and essential oils. Despite its reported health benefits, the presence of phenolic compounds at high content may affect the sensory characteristics of food, namely the color, flavor and also astringency. This is particularly important when developing a new food or drink, as it helps predict consumer acceptance regarding its flavor and taste [9].

According to the SpiritsEUROPE Association, the sales of spirits and distillate beverages in the European Union reached a value of 25 million hectoliters in 2021, adding up to €37 billion worth [10]. This sector is known for its high level of quality and product diversity, with 44 categories of spirits (Reg. (UE) 2019/787) [11]. Spirits constitute a type of distillate with a minimum alcohol content of 37.5% (*v*/*v*), known for its high volatile content. The aging of spirits in wooden barrels, traditionally made of oak, contributes to an effective improvement of their sensory profile, reducing the volatile content and bitterness and consequently making them more appealing to the consumer. However, the growing demand for oak wood has caused a noticeable increase in costs due to the limited availability of materials and an ecological impact. Thus, the use of other alternatives, such as wood chips or nutshells, may be an interesting option to improve the aging process of the spirit and its phenolics content.

Liqueurs, on the other hand, are a type of distillate with a minimum alcohol content of 15%, known for their sweetness and varied flavoring aromas, usually produced from spirits (aged or not). The sweetness is given by the addition of sugar or honey. The flavorings used for the production of liqueurs range from complex mixtures of fruits, aromatic herbs [12] and flowers [13]. Schuina et al. [14] successfully used *carqueja* (*Baccharis trimera*), originally produced in Brazil, as a total substitute for hops in the production of lager beer. The final product demonstrated excellent quality, with satisfactory physicochemical characteristics similar to the hop beer with good sensory acceptance. However, some authors highlight the potential use of this vegetable in the spirits department as a flavoring agent. Simões et al. [2] and Aires et al. [6] reported that hydroalcoholic extracts from *Pterospartum tridentatum* (*carqueja*) have a high concentration of flavanols, flavones and isoflavones, making this plant very attractive for spirits flavoring.

In the present study, a *carqueja* liqueur was produced to evaluate the influence of using aged wine spirits with almond shells and *carqueja* flowers on the physicochemical characteristics, phenolic composition and sensory characteristics of the developed product. The characterization and identification of the phenolic profile of the aged wine spirit as well as the liqueur allowed for the demonstration of new use and valorization of the almond shells during aging, as well as the use of the medicinal flower *carqueja* as a flavoring agent in the spirits market.

## 2. Results & Discussion

### 2.1. Physicochemical Analysis of the Spirits and the Liqueur

Table 1 shows the physicochemical characteristics of the wine spirits (WS and WSA) and liqueur (CL), pH, alcohol content, dry extract, color characteristics, acidity (total, fixed, and volatile) and viscosity. The differences between CL and the two wine spirits (WS and WSA) were clearly evident. Only in the case of total acidity did CL values not show statistical differences with those of WS without aging. These differences are due, to some extent, to the dilution of CL with sugar syrup and to the properties of the *carqueja* flower.

All spirits in Table 1 had very acidic pH levels with values between 2.09 ± 0.09 (WS) and 4.17 ± 0.01 (CL). Spirits and liqueurs naturally tend to have these pH values thanks to the high acid content present in these matrices [12,15,16]. The pH values also showed a tendency to increase slightly over time due to the evaporation of volatile acids during the aging process, which is also correlated with the stabilization of spirits, improving their organoleptic characteristics [17].

The alcohol content of the spirits remained almost constant during the aging process, from 53.1% in the WS to 51.2% in the WSA. The aging process was stable, and no evident loss of alcohol was demonstrated during the alternative aging process applied with almond shells. Furthermore, the possibility of air contact with the wine spirit did not contribute significantly to the secondary oxidations of this process. The alcohol content decreased considerably during CL production (to 18.3%) due to the dilution carried out in the process of adding sugar syrup.

The dry extract increased significantly during the aging process between WS and WSA and later to CL. The main difference between these results is that, for WSA, the dry extract increased due to the extraction of compounds from the almond shell, and for CL, a significant part of the extract comes from compounds from the *carqueja* flowers and, mainly, from the added sugar.

The increase in color intensity of the WSA, which reaches a value of 0.481 (Table 1), is solely due to the aging process with the almond shells. As far as CL color is concerned, this is also due to the *carqueja* extract reaching a maximum level of 1.509 (Table 1), with the luminosity decreasing to the lowest value of 89.54 and the a* and b* parameters reaching values that on the CIEL*a*b* color space correspond to yellow/orange color (as shown in Figure 1). Luís et al. [18] concluded that ethanolic solvents are more effective in the extraction of carotenoids such as beta-carotenes, which are known to be the pigments responsible for the yellow color of foods, making them more attractive to consumers.

The total acidity remained relatively constant during the aging process. On the contrary, the fixed acidity and volatile acidity showed a negative correlation between them, where the fixed acidity increased during the aging process, from 282.40 mg acetic acid/L at WS to the highest value at CL of 759.20 mg acetic acid/L (Table 1), due to the extraction of more stable compounds (such as phenols). As for volatile acidity, it decreased from its highest value of 856.00 mg of acetic acid/L in WS to 352.00 mg of acetic acid/L in CL (Table 1). The difference between the fixed and volatile acidity of WS and WSA can be explained by the effect of the different components of wood and almond shell. The increase in fixed acidity in CL may be due to the extraction of *carqueja* flower, and the decrease in volatile acidity may be an effect of the dilution of WSA relative to WS. Other works that have analyzed different types of liqueur tend to assess only its total acidity [12,15,16], which is only comparable if it is the same type of liqueur since the acidity of the liqueur is inherently correlated with the base used for its production. The evaluation of volatile and fixed acidity is not common in this type of product, but our results can correlate with the previous statement that during maturation, the volatile acids evaporate, decreasing the volatile acidity and extracting more stable compounds while increasing the fixed acidity.

### 2.2. Total Phenolic Compounds, Flavonoids, and Tannins

As regards the *carqueja* extract (CE), an extraction yield of 18.7% was obtained, which is similar to a previous work carried out with aqueous extracts of *carqueja* from the same region (Gardunha), which presented extraction yields in the range of 17.2–18.1% [4,19]. Additionally, higher extraction yields (28.6%) were obtained for ethanolic extracts of species collected near our region (Serra Estrela, Portugal) [16]. This higher value can be justified by the fact that extraction conditions are inherently linked to the type of *carqueja* used, as well as its growing region, environment and microclimates, thus influencing the extraction yield and playing an important role in the development of certain compounds that may be more prone to extraction.

Observing Table 2, it can be concluded that the WS presented residual contents of phenols, flavonoids and tannins, whose origin might be related to the wine that was previously distilled. During the aging process of the WSA, the contents of phenols, flavonoids and tannins increased significantly. These compounds come from the almond shell itself since no other agent was present that could have contributed to these results, thus suggesting that almond shell contains significant phenolic content, as shown in some previous works [20,21,22,23]. Queirós et al. [20] and Prgomet et al. [22] are some of the few works similar to ours that show that flavonoids and tannins constitute an important part of the phenolic composition of almond shell. These compounds are known to be present during the aging and maturation of spirits and liqueurs, giving them a high antioxidant power. Therefore, future work should be focused on the composition and content of these substances to assess the suitability of almond shells for the aging of spirits or liqueurs, as well as their potential use in other food products.

The highest increase in phenols, flavonoids and tannins was observed during liqueur maturation (CL) due to the use of *carqueja* flower for flavoring the liqueur. In terms of phenol and flavonoid content in *carqueja*, it has already been reported that *carqueja* has a high phenol and flavonoid content [3,4,18,19], which corroborates this increase.

Luís et al. [18] even showed that ethanolic solvents seem to be better at extracting flavonoids from *carqueja* flowers than water, which is in line with the results of our work, both for the liqueur (8642.7 mg EACT/L) and the extract (7175.4 mg EACT/L). Furthermore, it is known that flavonoids are mainly responsible for the flavoring of food, which proves that *carqueja* can be an adequate flavoring agent for spirits and liqueurs. Several authors conclude that *carqueja* flowers have high antioxidant capacities [3,4,5,7,18], with Luís et al. [18] even demonstrating that ethanol is more effective in extracting phenols with higher antioxidant capacity.

When the tannin content of CL is analyzed, it can be seen that the spirit used for its production was not proficient in extracting such high concentrations as in the extracts, considering the dilution made by the syrup, while the extraction conditions of the CE proved to be more efficient. The matrix of the spirit is indeed more complex than the mixture of water and ethanol used for the CE; however, it must be taken into account that the WSA used for the liqueur had already been aged. Since the WSA already has a complex matrix composed of phenols, flavonoids and tannins, it made the spirit more stable and saturated for the extraction of further tannins. In addition, tannins are usually very stable, and some are very condensate, which makes them not as prone to extraction as other phenols and flavonoids. To our knowledge, there is only one work that quantifies the total amount of tannins in the *carqueja* extracts [24], reinforcing the need for further research that allows the isolation and quantification of these compounds, as they are one of the most important polyphenolics found in plants.

### 2.3. Analysis of the Phenolic Compounds by HPLC

Specific analysis and quantification of the phenolic profile were performed using an HPLC-LC-DAD-ESI/MSn. The detected compounds are shown in Table 3 with information on their retention time, the wavelength of maximum absorption, pseudomolecular ion, mass fragmentation and tentative identification. Table 4 presents the values that were quantified for the respective compounds summarized in Table 3. In Figure 2, it is presented an exemplificative phenolic profile of the CL sample.

There was an unknown compound (**Peak 3**) that was only detected in the WS and WSA; since it was not detected in the CL, it was determined it was probably a volatile compound from the spirit and therefore decided not to quantify it, since it did not present relevance in the analyze of the phenolic profile.

For the wine spirit (WS), no phenolic compounds were detected as expected since this type of spirit is known for the lack of phenols. As such, the residual content shown in Table 2 could be due to quantification errors since the spectrophotometric method used is known for wrongly quantifying copper as a phenol [25].

**Table 3 molecules-28-04455-t003:** Retention time (Rt), wavelength of maximum absorption, pseudomolecular ion, mass fragmentation and tentative identification of the phenolic compounds found in wine spirit (WS), wine spirit aged with almond shells (WSA), *carqueja* liquid extract (CE) and *carqueja* liqueur (CL).

Peak	Rt (min)	λmax (nm)	[M − H]^−^ (*m*/*z*)	MS^2^ (*m*/*z*)	Tentative Identification	Reference
**1**	4.82	291, sh339	465	447 (9), 375 (12), 357 (5), 345 (100), 327 (11), 317 (5), 167 (7)	Dihydroquercetin-*C*-hexoside	[7]
**2**	5.84	284, sh340	479	359 (100), 341 (5), 221 (5), 167 (5)	Myricetin-*C*-hexoside	[7]
**3**	6.68	292, sh338	-	-	Unknown compound	-
**4**	8.29	258/301	465	447 (9), 375 (13), 357 (5), 345 (100), 327 (15), 317 (5), 167 (7)	Dihydroquercetin-*C*-hexoside	[7]
**5**	11.54	367	413	311 (100), 269 (25)	Genistein derivative	DAD/MS
**6**	12.41	255/320	491	445 (10), 283 (100), 269 (60)	3′-Methoxy daidzin	[26]
**7**	13.36	261/320	431	311 (100), 283 (10)	Genistein-*C*-hexoside	DAD/MS
**8**	13.84	261/320	431	311 (100), 283 (12)	Genistein-*C*-hexoside	DAD/MS
**9**	16.4	352	609	301 (100)	Quercetin-*O*-deoxylhexosyl-hexoside	DAD/MS
**10**	16.55	357	609	301 (100)	Quercetin-*O*-deoxyhexosyl-hexoside	DAD/MS
**11**	17.25	354	463	301 (100)	Quercetin-*O*-hexoside	DAD/MS
**12**	17.56	353	463	301 (100)	Quercetin-*O*-hexoside	DAD/MS
**13**	18.33	260/329	463	301 (100)	Ellagic acid hexoside	DAD/MS
**14**	19.85	261/321	433	301 (100)	Ellagic acid pentoside	DAD/MS
**15**	21.5	260/322	431	311 (10), 269 (100)	Genistein 7-*O*-glucoside (Genistein)	Composto Padrão
**16**	22.92	256/320	505	459 (5), 297 (100), 282 (76)	Methylbiochanin A/methylprunetin *O*-hexoside	[7]
**17**	30.33	368	301	179 (100), 151 (78)	Quercetin	DAD/MS
**18**	31.82	260/320	491	445 (3), 283 (100)	Biochanin A *O*-hexoside	[7]
**19**	34.53	260/320	269	241 (4), 225 (6), 201 (5), 181 (2), 133 (7)	Genistein	DAD/MS
**20**	35.26	260/320	283	268 (100), 239 (7), 224 (5), 195 (2), 135 (2)	4′-*O*-Methylgenistein (biochanin A)	[7]
**21**	36.72	260/320	297	282 (100)	Methylbiochanin A/methylprunetin	[7]
**22**	44.33	260/320	283	268 (100)	7-*O*-Methylgenistein (prunetin)	[7]

**Table 4 molecules-28-04455-t004:** Phenolic compounds (mg compound/L) of the wine spirit (WS), wine spirit aged with almond shells (WSA), *carqueja* liquid extract (CE) and *carqueja* liqueur (CL) (Mean values ± standard deviation).

Compound	WS	WSA	CE	CL
Dihydroquercetin-*C*-hesoxide	nd	18.3 ± 0.2 a	68.4 ± 6.6 c	28.9 ± 0.5 b
Myricetin-*C*-hexoside	nd	nd	138.4 ± 11.0 *	67.9 ± 4.5 *
Unknown compound	nq	nq	nd	nd
Dihydroquercetin-*C*-hexoside	nd	7.0 ± 0.3 a	57.6 ± 14.3 c	16.3 ± 0.3 b
Genistein derivative	nd	nd	11.1 ± 0.5 b	2.5 ± 0.1 a
3′-Methoxy daidzin	nd	9.8 ± 0.1 *	134.9 ± 8.4 *	nd
Genistein-*C*-hexoside	nd	tr	23.8 ± 2.6 *	9.4 ± 0.2 *
Genistein-*C*-hexoside	nd	nd	22.2 ± 0.8	nd
Quercetin-*O*-deoxylhexosyl-hexoside	nd	nd	13.4 ± 1.3 *	6.2 ± 0.1 *
Quercetin-*O*-deoxyhexosyl-hexoside	nd	nd	24.7 ± 3.0 *	6.2 ± 0.2 *
Quercetin-*O*-hexoside	nd	nd	214.4 ± 14.3 *	13.6 ± 0.2 *
Quercetin-*O*-hexoside	nd	nd	109.3 ± 18.3 *	9.1 ± 0.1 *
Ellagic acid hexoside	nd	nd	151.8 ± 8.6	nd
Ellagic acid pentoside	nd	21.7 ± 0.04 b	36.9 ± 1.9 c	14.5 ± 0.1 b
Genistein 7-*O*-glucoside (Genistein)	nd	nd	19.5 ± 1.0	nd
Methylbiochanin A/methylprunetin *O*-hexoside	nd	nd	257.6 ± 33.6	nd
Quercetin	nd	6.3 ± 0.05 a	47.3 ± 3.9 b	6.6 ± 0.3 a
Biochanin A *O*-hexoside	nd	nd	201.4 ± 13.9	nd
Genistein	nd	nd	95.1 ± 6.6 *	0.1 ± 0.01 *
4′-*O*-Methylgenistein (biochanin A)	nd	nd	180.5 ± 58.5	tr
Methylbiochanin A/methylprunetin	nd	nd	137.8 ± 26.1	tr
7-*O*-Methylgenistein (prunetin)	nd	nd	146.3 ± 6.5	nd
Total isoflavones	nd	tr	1230.2 ± 139.2 *	12.0 ± 0.3 *
Total ellagic acid derivatives	nd	33.6 ± 0.04 b	188.7 ± 7.0 c	14.5 ± 0.10 a
Total flavonoids	nd	54.5 ± 0.10 a	673.6 ± 42.7 c	154.7 ± 4.2 b
Total Phenolic Compounds	nd	43.4 ± 0.04 a	2092.4 ± 176.5 c	181.2 ± 3.8 b

Terms: nd—not detected; nq—not quantifiable; tr—trace amounts. Biochanin A (y = 35,267x + 352,964, **peaks 16**, **18**, **20**, and **21**); Daidzin (y = 27,652x + 29,187, **peak 6**); Ellagic acid (y = 26719x – 317,255, **peaks 13** and **14**); Genistein (y = 64,642x + 187,360, **peaks 5**, **7**, **8**, **15**, **19**, and **22**); Quercetin-3-*O*-glucoside (y = 34,843x − 160,173, **peaks 1**, **2**, **4**, **9**, **10**, **11**, **12**, and **17**). a–c Different lower-case letters in the same row indicate statistically significant differences (*p* < 0.05). * *t*-test student (*p* > 0.001).

Regarding the phenolic profile of the almond shells, Prgomet et al. [22] proposed a chromatogram for the evaluation of almond shells that shows to be richer and more complex than the other studied components (skins and hulls); however, the authors could not identify any phenolic compound. This work appears to be the first to successfully identify and quantify six phenolic compounds from the almond shells, corresponding to taxifolin (**peaks 1** and **4**), isoflavone daidzein (**peak 6**) and the derivates of quercetin (peak 17), ellagic acid (**peak 14**) and genistein (**peak 7**). All these compounds have been previously identified in other parts of the almond tree, such as stems, leaves or kernels [27,28,29].

*Carqueja* liquid extract (CE) was revealed to have the most complex profile of all the samples. In fact, 21 out of 22 peaks were observed; the only one that was not present was **peak 3**, previously described as a volatile compound of WS and WSA. As for the liqueur, it was only possible to identify 14 of the 21 compounds detected in the extracts and quantify 12 of them, thus reinforcing its complex matrix. Additionally, important phenols known for their antioxidant power, such as genistein, prunetin and quercetins, have also been detected and quantified in significant concentrations [5].

Based on the literature, 21 of the identified compounds have been previously reported in the same forms or as derivates in several works. **Peaks 1** and **4** were identified as Dihydroquercetin-C-hesoxide, also known as Taxifolin-3-glucoside, a taxifolin derivate [6,7,19]; **peak 2** as a myricetin derivate [7,19,30]; **peaks 5**, **7** and **8** were identified as genistein derivates, and **peak 19** as genistein [6,7,19,30]; **peak 15** was identified as genistin [5,6,7,19,30]; **peak 17** was identified as quercetin and **peaks 9**, **10**, **11** and **12** as quercetins derivates [5,6,7,19,24,30]; Biochanin A was identified in **peak 18**, and a derivate was found in **peak 20** [6,7]; Ellagic acid was also identified in the hexoside form in **peak 13** and pentoside form in **peak 14** [24]. Prunetin was identified in **peak 22** [5,7,30], and **peak 16** and **peak 21** were identified as possibly methylbiochanin A or methylprunetin [7]. **Peak 6** was identified as a daidzein derivate—daidzein is an isoflavone known to be present in other medicinal plants [26] and in *carqueja* roots [2], but this is the first time it has been identified in *carqueja* flowers. This work demarks itself for being the first to detect and quantify prunetin, daidzein, biochanin-A and ellagic acid in the *carqueja* subspecies from the Gardunha region (Portugal) [19].

### 2.4. Sensorial Analysis

Sensory analysis is essential to perceive the quality and acceptability of a product from the consumer’s point of view [17]. The results of the sensory analysis and the data regarding purchase intention are presented in Figure 3.

In the sensory analysis, we sought to evaluate the perception of the “alcoholic content” by the consumer since the WSA, used in the production of CL and analyzed in the trial, was known for its higher alcoholic content (53.12%) when compared to the liqueurs CL (18.3%) and AL (15%). Thus, the WSA exhibited a higher perception rate (6.43), and the CL (3.98) and CE (4.31) liqueurs revealed similar rates. It is important to note that, despite CL having a higher alcoholic content than AL, it showed a lower perception rate than AL, strongly justified by the balance between sweetness and bitterness. Sweeter drinks are generally perceived to have lower alcohol content, even though this may not be the case. On the contrary, bitterness is more associated with higher alcohol content. Thus, the sample with the highest bitterness rate is WSA, followed by AL, and lastly, CL. As expected, for sweetness, the opposite is found.

In terms of color, all samples perceived the same tonality (color intensity, L*, a* and b*) of golden/topaz, as shown in Table 1. Noticeable differences were observed in terms of intensity. In the sensory analysis, among the three samples, the least appreciated was AL (with weaker intensity), and the most appreciated was WSA. The *carqueja* liqueur (CL) revealed an intense hue and some opacity, which seems to be a result of the high extraction of carotenoids of the *carqueja* flower promoted by ethanol [18]. However, the opacity of this sample became an obstacle to a better appreciation. A further filtration step could improve the clarity of the product and, therefore, its appreciation.

Regarding aroma, flavor and global appreciation, CL presented the best score with values close to or higher than 6. This can be explained by the fact that *carqueja* flower is very aromatic, with highly appreciated aromas and flavors. WSA, as expected, presented the lowest appreciation and the highest astringency since it does not contain sugar or *carqueja* flower compounds in its composition. Interestingly, and despite containing *carqueja* flower, AL was not as well appreciated as CL since it does not present high complexity in aroma and flavor, mainly due to the combination of unaged wine spirit with an infusion of *carqueja* flower. This infusion does not seem to have a high aromatic power, which is in contrast to the slow and direct extraction used for CL. Moreover, unaged wine spirits are not as highly regarded as the aged ones. Overall, CL was the most appreciated spirit, with a rating of (6.12) against AL (5.05).

Regarding the product purchase intentions, panelists prioritized the CL over the AL, as for the CL, they revealed a high purchase intention of 94.7% for the options of “Would buy” and “Probably would buy”, while for AL, it was 71.9%. It is also important to note that CL was the only sample that showed a 0% rate for the option “Wouldn’t buy”, reinforcing its acceptance among consumers.

## 3. Materials and Methods

### 3.1. Materials and Samples for Analysis

The flowers of *Pterospartum tridentatum* (*carqueja* flowers) were harvested in Serra da Gardunha (Fundão, Portugal) during the flowering season (late May). The flowers were naturally air-dried at room temperature (15–25 °C), in the absence of light, for 3 months. To study the extraction process and the chemical composition of *carqueja* liqueur (CL), produced from wine spirit aged with almond shells (WSA), wine spirit (WS) and ethanolic extract of *carqueja* flowers (CE) were used as control samples.

The WS was obtained from the distillation of wine in the winery of the Agriculture School of Polytechnic Institute of Coimbra.

To obtain the WSA, the WS was aged for four months with almond shells using a faster aging method. The proportion of almond shells to wine spirits was 50 mg of shells/mL. Before aging, almond shells were previously separated from the fruits, purchased in a local market in Coimbra (Portugal), and toasted in an oven for 60 min at 200 °C to simulate the toasting process of the wood chips used in the aging of the spirits.

The *carqueja* liqueur (CL) was produced with 1.5 L of aged wine spirit (WSA) combined with 300 g of dried *carqueja* flowers, which were left to steep for 1 month in the liqueur. The liquid was then filtered, and 2 L of syrup (sugar:water ratio of 1:1 (*w*/*v*)) was added. The mixture was left to mature for 10 days before analysis.

An ethanolic extract of *carqueja* flower (CE) was also prepared for comparison with the wine spirit extraction. An ethanolic solution with the same alcohol content as the wine spirit (53%, *v*/*v*) was used as the extraction solvent for *carqueja* flowers at a concentration of 0.2 g/mL for 180 min at room temperature with continuous stirring at 180 rpm. The CE was then filtered and stored under refrigeration conditions before analysis.

An artisanal liqueur (AL) with an alcohol content of 15% was purchased directly from the producer located in the Castro Daire region (Portugal). It was used for comparison with wine spirit aged with WSA and CL during the sensory analysis.

All chemicals were of analytical grade and were used as received without further purification. All aqueous solutions were prepared with distilled or deionized water.

### 3.2. Physicochemical Analysis of Spirits and Liqueur

The total dry extract, pH and acidity (total, fixed and volatile) of WS, WSA and CL were performed according to the methods OIV-MA-BS-09, OIV-MA-BS-13 and OIV-MA-BS-12, respectively, of the Compendium of International Methods of Analysis of Spirituous Beverages of Vitivinicultural Origin [31]. For the liqueur (CL), a prior dilution of 1/100 was made before the determination of the total dry extract.

The chromatic characteristics of the beverages were analyzed according to the OIV-MA-BS-27 method [31] using a CHROMA METER CT-320 (MINOLTA) and a 2 mm cell. The color coordinates of L*, a* and b* were determined. Color intensity was determined following the OIV-MA-BS-26 method [31] using a T80+ UV/VIS Spectrometer (PG Instruments Ltd., Lutterworth, UK) at 445 nm with a 1 cm cell. Wine spirit (WS) was used as blank sample for both methods.

The alcohol content by volume was evaluated by hydrometry for wine spirit (WS) and the aged wine spirit (WSA). The alcohol content of the liqueur (CL) was assessed by Acal ebulliometer (model 2, Porto) after dilution with water in a 1:1 ratio prior to analysis.

The liqueur viscosity was measured with a rotational viscometer (BROOKFIELD AMETEK–DV2T) using RV-1 spindle, 150 rpm, average torque of 94.2% and accuracy of 26.67. The temperature during the test was 19 ± 0.1 °C. All analyses were performed in triplicate.

### 3.3. Determination of Total Phenolic Compounds

Total phenolic compounds (TPC) were quantified for all samples by the Folin–Ciocalteu method [32]. The liquid samples were properly diluted with 70:30 methanol to a final volume of 200 µL before being mixed with 200 µL of the Folin–Ciocalteu solution and 1.6 mL of 5% Na_2_CO_3_. The samples were left in a 40 °C water bath for 20 min, then the absorbance was read at 750 nm against the blank (70:30 methanol). The calibration curve was prepared with solutions of gallic acid (Sigma-Aldrich, St. Louis, MO, USA, CAS:149-91-7) at concentrations of 0.00 to 0.05 mg/mL, and the results expressed in mg of gallic acid equivalents (mg GAE) per mL of sample. The analysis was performed in triplicate.

### 3.4. Determination of Total Flavonoids

Total flavonoids (TF) were determined according to the method described by Kim et al. [33]. The samples were properly diluted in a test tube with 50:50 methanol to a final volume of 1 mL, then 4 mL of bi-distilled water was added before being mixed with 0.3 mL of 5% NaNO_2_, 0.3 mL of 10% AlCl_3_, 2 mL of NaOH 1 M and 2.4 mL of bi-distilled water. Absorbance was read at 510 nm on a T80+ UV/VIS Spectrometer (PG Instruments Ltd.). The calibration curve was prepared with solutions of epicatechin (Sigma-Aldrich; CAS:225937-10-0, ≥98%) with concentrations of 0.0 to 0.2 mg/mL, and the results were expressed in mg of epicatechin equivalents (mg ECAT) per mL of sample. The analysis was performed in triplicate.

### 3.5. Determination of Tannins

Tannins (Tan) was established by the titrimetric method AOAC 30.018 [34] and using the guidelines of Atanassova and Christova-Bagdassarian [35]. In total, 25 mL of the samples were measured into a 1 L Erlenmeyer flash and combined with 25 mL of indigo solution and 750 mL of distilled water. The titration was performed with 0.1 N aqueous KMnO_4_ solution until the blue color changed to green, then one drop at a time was added until the solution turned golden yellow. The blank sample consisted of the titration of a mixture of 25 mL of indigo carmine solution and 750 mL of distilled water. The results were expressed in mg tannins equivalents (ET) per g of sample. The analysis was performed in triplicate.

### 3.6. Analysis of the Phenolic Compounds by HPLC

The phenolic profile of the liquid samples was determined by LC-DAD-ESI/MSn (Dionex Ultimate 3000 UPLC, Thermo Scientific, San Jose, CA, USA). Phenolic compounds were separated and identified as previously described by Bessada et al. [36]. The obtained liquid extracts were diluted at a concentration of 50 mg of the liquid extract/mL with a mixture of ethanol:water (80:20, *v*/*v*). A double online detection was performed using a DAD (280, 330, and 370 nm as preferred wavelengths) and a mass spectrometer (MS). The chromatographic separation was achieved using a Waters Spherisorb S3 ODS-2 C_18_ (3 µm, 4.6 mm × 150 mm, Waters, Milford, MA, USA) column thermostatted at 35 °C. The solvents used were 0.1% formic acid in water and acetonitrile. The MS detection was performed in negative mode, using a Linear Ion Trap LTQ XL mass spectrometer (Thermo Finnigan, San Jose, CA, USA) equipped with an ESI source. The identification of phenolic compounds was performed based on their chromatographic behavior and UV-vis and mass spectra by comparison with standard compounds, when available, and data reported in the literature to attempt identification.

Data acquisition was carried out using the Xcalibur^®^ data system (Thermo Finnigan, San Jose, CA, USA). For quantitative analysis, a calibration curve for each available phenolic standard was constructed based on the UV-vis signal. For identified phenolic compounds for which there was no commercial standard available, quantification was performed using the calibration curve of the most similar standard available. The results were expressed as mg/L of the sample.

### 3.7. Sensorial Analysis

For the sensory analysis, three samples were analyzed: wine spirit aged with almond shells (WSA), *carqueja* liqueur (CL) and an artisanal *carqueja* liqueur (AL). The test was carried out in the sensory room equipped with 9 specific cabinets at the Agriculture School (Coimbra, Portugal) by an untrained panel due to the restrictions imposed by the COVID-19 pandemic. A total of 57 subjects, aged between 18 and 61 years, were asked to rate the samples on color, aroma, flavour and overall appreciation, using a hedonic scale from 1 to 7 (where 1—dislike and 7—liked too much). To assess the perception of intensity of sweetness, bitterness, and alcohol content, the 7-point hedonic scale was graded from 1—not intense to 7—very intense. At the end of the sensory analysis, the purchase intention was also questioned.

The samples were provided to the panelists at the same time in plastic cups (15 mL) covered to avoid evaporation and blending of aromas. A napkin and a glass of drinking water were also provided to clean the palate between samples. The test was performed during the morning to maintain a better state of satiety and better adaptation to the sense of smell [37].

### 3.8. Statistical Analysis

The results of the physicochemical analyses, total phenolic compounds, total flavonoids, tannins and phenolic compounds by HPLC, were presented as mean values and standard deviation. One-way ANOVA tests, included in StatSoft Statistica 10.0 (StatSoft Inc., Tulsa, OK, USA), were performed to compare the means of the physicochemical properties and the attributes used for sensory evaluation. The Tukey HSD post hoc test, with a 95% confidence level, was applied to assess differences between samples.

## 4. Conclusions

As far as we know, this work presents the only detailed study of liqueur production using *carqueja* flowers. Comprehensive characterization of the phenolic compounds present in the CE was achieved by identifying 21 phenolic compounds under study by HPLC-DAD-ESI/MSn analysis.

Some specific phenols found, such as genistein, are responsible for providing an anti-inflammatory response in the wound healing process and quercetins, as they disrupt the synthesis of nucleic acids of the cytoplasmic membrane and the energy metabolism of bacteria. Furthermore, it is also important to note that the concentration used in our work (0.2 mg/mL) is below the minimum concentration known to produce adverse cytotoxic effects on human health (0.375 mg/mL) [3]. In addition, some *carqueja* plants may exhibit the presence of sissotrin, depending on the region and environment in which they grow. This compound is specifically known to produce an anti-effect to that observed by isoquercitrin by impairing glucose tolerance [30]. Sissotrin was not identified in this work in LC or CE. However, it was identified in another work with aqueous extracts of *carqueja* also from the Gardunha region [19].

Regarding the consumer’s preferences, the liqueurs (CL and AL) showed a higher appreciation and purchase intention, rather than the aged spirit (WSA) due to its specific sensory characteristics, such as sweetness and smoothness. The CL liqueur presents a complex and distinguishable sensory profile while providing health benefits. Furthermore, the valorization of this flower is also important for the enrichment of its regions of origin, such as Beira Interior and Trás-os-Montes (Portugal). We consider that the application of *carqueja* flower in food product development should be further explored in an industrial-scale context with the implementation of a proven concept to validate the technology and the final product composition that may be of particular interest to liqueur producers.

## Figures and Tables

**Figure 1 molecules-28-04455-f001:**
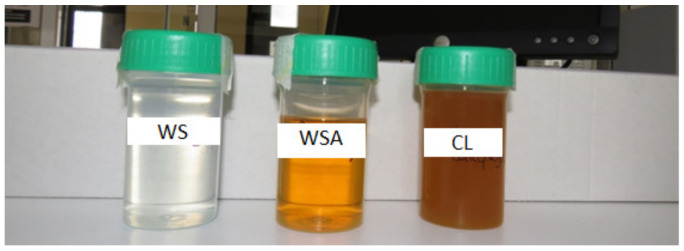
Color of wine spirit (WS), aged wine spirit with almond shells (WSA), and liqueur (CL).

**Figure 2 molecules-28-04455-f002:**
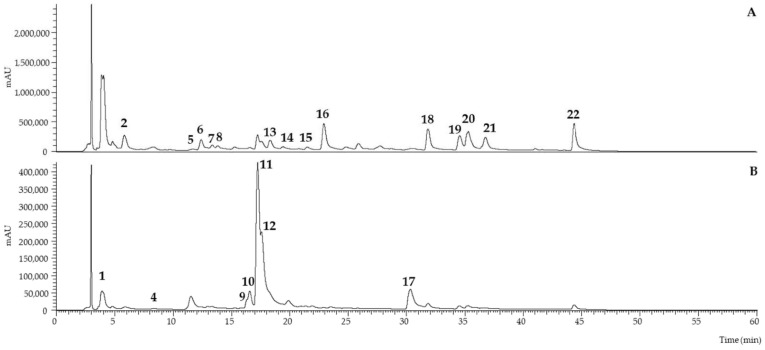
Exemplificative phenolic profile of CL sample recorded at 280 nm (**A**) and 370 nm (**B**).

**Figure 3 molecules-28-04455-f003:**
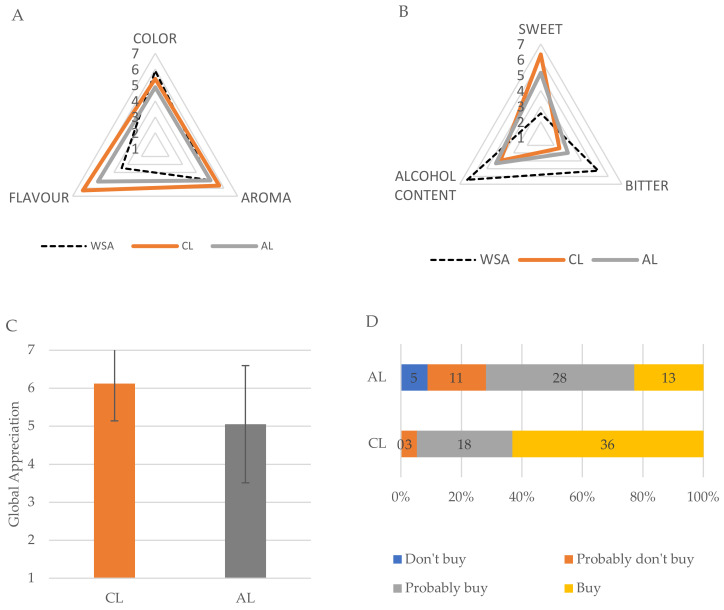
Sensory analysis of the wine spirit aged with almond shells (WSA), *carqueja* liqueur (CL) and artisanal *carqueja* liqueur (AL). (**A**) Color, aroma and flavour (hedonic scale: 1—dislike; 7—liked too much). (**B**) Perception of the intensity of sweetness, bitterness and alcohol content (hedonic scale: 1—not intense; 7—very intense). (**C**) Global appreciation of the liqueurs. (**D**) *Carqueja* liqueurs purchase intention.

**Table 1 molecules-28-04455-t001:** pH, alcohol content, dry extract, color characteristics, acidity and viscosity of the wine spirit (WS), wine spirit aged with almond shells (WSA) and *carqueja* liqueur (CL). (Mean values ± Standard deviation).

Parameter	WS	WSA	CL	*p*-Value
pH	2.09 ± 0.09 a	3.29 ± 0.01 b	4.17 ± 0.01 c	*p* < 0.001
Alcohol content (%, vol)	53.12 ± 0.38	51.57 ± 0.06	18.33 ± 0.23 a	*p* < 0.001
Dry extract (g/100 mL)	0.034 ± 0.280 a	0.248 ± 0.036 b	55.170 ± 2.832 c	*p* < 0.001
Color intensity (Abs_445 nm_)	0.095 ± 0.005 a	0.481 ± 0.002 b	1.509 ± 0.011 c	*p* < 0.001
Color				
L*	100.01 ± 0.00 c	98.57 ± 0.01 b	89.54 ± 0.27 a	*p* < 0.001
a*	−5.25 ± 0.02 a	−6.18 ± 0.01 b	−6.58 ± 0.07 c	*p* < 0.001
b*	5.42 ± 0.01 a	12.73 ± 0.01 b	23.47 ± 0.10 c	*p* < 0.001
Acidity (mg acetic acid/L)				
Total	1138.40 ± 19.4 a,b	1189.60 ± 1.39 b	1111.20 ± 35.52 a	*p* < 0.05
Fixed	282.40 ± 8.43 a	409.60 ± 12.32 b	759.20 ± 1.39 c	*p* < 0.001
Volatile	856.00 ± 12.32 c	780.00 ± 12.70 b	352.00 ± 36.90 a **	*p* < 0.05 ** *p* < 0.001
Viscosity (cP)	-	-	2.511 ± 0.002	-

a–c Different lower-case letters in the same row indicate statistically significant differences at (*p* < 0.001). ** Indicate statistically significant differences at (*p* < 0.05).

**Table 2 molecules-28-04455-t002:** Total phenolic compounds (TPC), total flavonoids (TF) and tannins (Tan) of wine spirit (WS), wine spirit aged with almond shells (WSA), *carqueja* liquid extract (CE) and *carqueja* liqueur (CL) (Mean values ± Standard deviation).

Parameter	WS	WSA	CE *	CL
TPC (mg GAE/L)	23.03 ± 0.11 a	211.29 ± 2.18 b	1485.41± 60.69 c	1640.92 ± 3.36 b
TF (mg EACT/L)	49.13 ± 0.10 a	601.60 ± 10.10 b	7175.44 ± 70.16 c	8642.68 ± 69.28 d
Tan (mg ET/100 g)	0.61 ± 0.00 a	30.60 ± 1.22 b	811.51 ± 12.32 d	116.66 ± 2.33 c

* *Carqueja* liquid extract (CE) with the same dilution used in the *carqueja* liqueur (CL). a–d Different lower-case letters in the same row indicate statistically significant differences (*p* < 0.05).

## Data Availability

The data generated with this study are present in the results section of the present manuscript.

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
