# Peer review of "Pterospartum tridentatum Liqueur Using Spirits Aged with Almond Shells: Chemical Characterization and Phenolic Profile"

_molecules, 2023, doi:10.3390/molecules28114455_

Round 1
Reviewer 1 Report
General Comments.
The aim of the paper titled: Pterospartum tridentatum Liqueur using Spirits Aged with almond Shells: Chemical Characterization and Phenolic Profile, is the characterization and identification of the phenolic profile of the aged wine spirit as well as the liqueur, allowed to demonstrate a new use and valorization of almond shells during aging as well as the use of the medicinal flower carqueja as a flavoring agent in the spirits market.
In my opinion the work is interesting, and the objectives are solved but not well defined in the introduction. I will make some recommendations and criticisms that can help improve the article to the authors.
1) In page 1, line 22, the authors write: bioative and phenolic, must be bioactive and phenolic.
2) In page 2, line 53, the authors write: According to the SpiritsEUROPE Association (spiritsEurope, 2021), this reference must be better specified in the References section. Check the number of 5 million hectolitres indicated in page 2 line 54.
3) In page 2, line 76-79, the authors write: In this latter study, a carqueja liqueur was produced in order to evaluate the influence of using aged based wine spirit along with almond shells, and carqueja flowers in the physicochemical characteristics, phenolic composition, and sensory characteristics of the developed product. It’s not clear if this sentence is for reference 6. If it’s for the work, with the next sentence (lines 79-83) must be better written the objective of the work.
4) In page 4, lines 335-336, the authors write: Total dry extract, pH and acidity (total, fixed and volatile) of WS, WSA, and CL were performed according to the methods OIV-MA-BS-09, OIV-MA-BS-13 and OIV-MA-BS-12. The OIV-MA-BS-09 is for an extract content less than 15 g/L. The authors have made some modifications of this method? They can explain it.
5) In lines 96-114, there are obvious comments about the diminutions of the alcoholic strength between the spirits and the liqueur, also with the increase of the extract. The phrase in line 113: “… the sugar that was added in the syrup step (approximately 500 g/L)” is not much technical. When someone produce a liqueur, the criterion is to define an alcoholic strength and a sugar content. It seems that the authors have prepared a liqueur without any criteria. I believe that in the materials section and here the authors must define better this part and justify the characteristic of the liqueur (CL).
6) In lines 115-121, the increase of colour between WSA and CL is not justified by the aging. These two products are different, CL has carqueja extract, while WSA the colour is due only to aging in almond shell. This part must be better explained and justified (Including the corresponding part in chapter 3, lines 8320-323).
7) In lines 329-330, the characteristics of AL must be better defined (alcoholic strength, sugar content, pH, …)
8) The explanation of acidity variation (fixed and volatile) with the aging in lines 124-136 in my opinion is not correct, the difference between WS and WSA can be explained by the effect of the wood components, and in the liqueur the increase of fixed acidity can be due to the carqueja extraction, and the diminution of volatile acidity can be effect of WSA dilution respect WS.
9) Section 2.1. Total phenolic compounds, flavonoids and tannins. I’m agree with this part of the paper.
10) In section 2. Results and discussion, all the other subsections are 2.1. Revise.
11) In line 140 the authors write: “… that presented extraction yields of 17.2%, 17.3% and 18.1% ...” Why three different values? Explain it.
12) In Table 2 (line 160) indicated TCP, but in the Table is TFC. Correct the mistake.
13) In line 172 the value indicated of 16742.7 mg EACT/ L is not the same of the Table 2.
14) In line 193 table 3 must be Table 3
15) In lines 197-198 indicate that the pic 3 is only identified in WS and WSA., but in Table 4 for WSA is indicated as not detected. Revise.
16) I recommend that the authors put the names of the compounds in Table 4.
17) Lines 249-251 the authors references Figure 2 and Figure 3, but in the paper is only figure 2, Correct the mistake.
18) In line 255 the authors write: … (18.3%) and CE (15%)… The acronym CE correspond to Carqueja alcoholic extract? Must be AL? Revise these mistakes in this section.
19) Regarding the explanation of lines 256-258 of the perception, it is necessary to know the sugar content of the commercial liqueur (AL). Because the sweetness of CL corresponds more to a cream than to a liqueur.
20) In my opinion the sensorial section can be reduced, because there are some discussion that are repeated.
21) The section of Conclusions is too long, it seems more like a discussion chapter than the conclusions of the work. It should be significantly reduced and focus on the main conclusions of the work.
In summary, I believe that the paper is interesting in some respects but must be improved notably to be publishable. Additionally, can be interesting for the liqueur’s producers.
Reviewer 2 Report
manuscript deal with chemical characterization and phenolic profile of pterospartum tridentatum liqueur.
Overall, the manuscript is well-organized and written correctly.
I have some comment on this work that have to be considered:
There is a lack of information about HPLC elution, e.g. column, mobile phase, type of elution ect.
Table 3, some of the lambda max are not in the Vis region, please correct the table caption.
Table 3 For some compounds only one mass (m/z) is provided, why? Additionally, is the same one for different compounds?
Why does one compound provide two peaks, or, why two peaks are identified as one compound?
An exemplary chromatogram may be useful.
How the quantitation of phenolic compounds was performed? Was the matrix effect evaluated? More information is necessary.
Round 2
Reviewer 2 Report
The manuscript was successfully corrected. The Author's responses to my comments are satisfactory.
I can recommend the manuscript for publication in its current form.